# Space Colonization and Exonationalism: On the Future of Humanity and Anthropology

Jack David Eller

Global Center for Religion Research, Denver, CO 80202, USA; david.eller.anthropology@gmail.com

**Abstract:** First anthropology became unbound from "the village", then from the single site, and gradually from the physical site altogether. As humans resume their push into space, anthropology is set to become unbound from the earth itself. This essay considers what the discipline has offered and can offer toward understanding the present and future of space colonization. It begins by examining the surprisingly long and productive history of anthropology's engagement with the subject, going back at least to the 1950s. Then it surveys current analysis of law, sovereignty, and nationalism in space, which largely imagines law and identity in off-earth settlements as more-or-less direct extensions or transfers of earth law and identity; in other words, space settlers will remain affiliated with and loyal to their source countries (or companies). However, taking seriously the analogy of terran migration and colonialism, where colonies developed distinct and separatist identities, the essay predicts the emergence of exonationalism, in which over generations colonists will invent new identities and shift their affiliations to their non-terran homes and ultimately seek independence from the earth. The essay concludes with reflections on how the settlement of space, still a distant goal, will reshape our definition of the human and therefore the practice of anthropology as the science of human diversity.

**Keywords:** colonization ethnogenesis; exonationalism; law; migration; space

## 1. Introduction

In Ursula Le Guin's 1974 novel *The Dispossessed*, a contingent of rebels and idealists had left the planet Anarres some two hundred years previously to colonize its moon or sister-planet, Urras. As followers of a philosopher-prophet named Laia Odo, the founder of Odonianism, the settlers established a socialist/anarchist society on their harsh world, with collective labor and without private property, in contrast to the capitalist (and in their minds corrupt) societies on the old world. In fact, the Urrasti not only regarded Urras as their true home but maintained minimal and tense relations with Anarres, forbidding Annarans to set foot on their planet outside of limited landing and trade zones and likewise prohibiting their own people from visiting the rival planet. Any citizen of Urras who ventured to Annares was labeled a traitor, suspected of sharing the planet's "propertarian" values and banned from return. Even communication and knowledge-exchange between the two peoples were restricted.

Although fictional, the narrative depicts a realistic relationship between opposing human worlds, based on the history of migration and colonization and humanity's demonstrated will to differentiate. Even in non-migratory situations, humans have often felt that members of the group who crossed cultural boundaries (say, nineteenth-century American settlers who chose to live among Native peoples, twentieth-century Americans who traveled to and spent time in the Soviet Union, or twenty-first-century Westerners who joined Islamist groups, if only temporarily) were no longer trustworthy, perhaps no longer "one of us". This attitude is likely to be much more pronounced and acute between space settlements and the earth, or between different space settlements, where distance will severely constrain contact between far-flung populations.

Anthropologists and other scholars, especially sociologists and political scientists, have already begun to ponder the political aspects of space colonization, mainly in the form of the extension of terran legal codes, nationalisms, and sovereignties to non-terran sites, the formulation of "space treaties" and "space law", and the potential political arrangements on off-earth settlements. Those are valid concerns, but they overlook another likely if not inevitable outcome, based on our experience of population movement and cultural-political evolution, namely, the development of separatist identities and movements in the colonies and the emergence of competition and hostility between, and the pursuit of independence for, those colonies. Taking seriously the notion of, and our knowledge of, colonial processes, this article explores the predictable rise of "exonationalism" in space, which means more than simply the transfer of earthly national identities and affiliations but the formation of unique local ones, the specifics of which we cannot foretell but the basic outlines of which we can anticipate confidently.

## 2. The Anthropology of Space (So Far)

In the third decade of the third millennium of the Common Era, and more than fifty years after mankind first landed on the moon, there may be some who still believe that anthropology has no business in space. I mean that in two senses—first, that anthropology as the "study of humanity" is inherently earth-bound, and second, that there is no business for anthropology to conduct in space, as there are no "sites" to perform participant observation and to compose ethnography in space. The former objection can be countered rather easily: where humans go, anthropology is free, indeed called, to follow; the fact that humans have been earth-bound *until now* is not a defining feature of anthropology. Actually, anthropology has been preparing for some time now for what scholars have called space anthropology [1] or extraterrestrial anthropology [2], at least since George Marcus advocated for multi-sited ethnography and its obligation "to examine the circulation of cultural meanings, objects, and identities in diffuse time-space" [3] (p. 96). In a concrete way, space settlements are only an expansion—if a dramatic expansion—of the "chains, paths, threads, conjunctions, or juxtapositions of locations in which the ethnographer establishes some form of literal, physical presence" (p. 105). It is not facetious to assert that Marcus' "ethnography in/of the world" takes on new dimensions as the human world grows—or multiplies and diverges—into human *worlds*.

The latter objection, that space offers no sites for ethnographic investigation, is half trivial and half false. Anthropology has long since transcended the "physical site" limitations of traditional fieldwork, once again readying itself for space research by penetrating into terrestrial "virtual spaces" such as cyberspace (e.g., internet communities, online gaming groups, etc.). However, anthropologists and allied researchers, particularly sociologists and archaeologists, have already found actual physical sites in which to practice the social scientific trade. One obvious option is the International Space Station, where art scholar Wendy Salmond collaborated with archaeologists Justin Walsh and Alice Gorman to examine (admittedly, from a distance) the material culture of Russian cosmonauts who carried and displayed religious icons onboard the station, creating "an economy of belief that spans Earth and space" [4] (p. 1). In an interview by Megan Gannon [5] on space archaeology and astronaut culture, Walsh and Gorman described a plan to assemble databases of all the objects taken to the station and "to create a 4D digital model of the space lab" in order to "understand patterns of life on the station in the same way that they might when looking at an archaeological site on Earth".

More commonly, and consistent with anthropology's contemporary interest in science and scientific practices in the laboratory, numerous anthropologists have conducted fieldwork within the earth-based space industry. Douglas Vakoch and various co-authors [6,7] have made the SETI program a subject of research, asking how anthropologists can contribute to the interpretation of extraterrestrial messages and the facilitation of extraterrestrial contact if and when it occurs. Anthropologists have also worked their way into various terrestrial simulations of future human space habitations, including Lisa Messeri's [8]

ethnography of the Mars Desert Research Station in Utah (in a book which also features a Mars mapping project, the telescopes of the Cerro Tololo Inter-American Observatory in Chile, and a search for exoplanets) and Savannah Mandel's [9] study of Spaceport America in New Mexico, which leads her to prophesize that space settlement will not solve humanity's inequality problem but intensify it (for a list of anthropologists and other scholars participating in social studies of outer space, see https://ssosnetwork.org/members, accessed on 14 September 2022) Finally, David Valentine, individually [10] and with colleagues Valeria Olson and Debbora Battaglia [11], has documented the burgeoning business of NewSpace, "the collective moniker for entrepreneurial space enterprises" such as Elon Musk's SpaceX and Jeff Bezos' Blue Origin that are "actively designing, building and launching space-faring vehicles and machines. From plans to beam solar power from space and mine asteroids, to visions of space tourism and moon colonies, effort and capital are being poured into these endeavors"; interestingly, though, Valentine, Olson, and Battaglia note that "NewSpacers are far more likely to talk about the deep future of human sociality in space than about the technology, costs or the risks of getting there" (p. 11).

It is worth noting that the contributions of anthropology to space programs substantially precede these academic activities. As early as 1957, rocket scientist Krafft Ehricke published "The Anthropology of Astronautics", and Margaret Mead and Rhoda Métraux were part of the interdisciplinary 1950s "Project Man in Space" team, producing the 1958 volume *Man in Space: A Tool and Program for the Study of Social Change*. In 1974 the American Anthropological Association held a symposium to discuss issues of space settlement, and Jack Stuster's consulting firm, Anacapa Sciences (anacapasciences.com. accessed on 14 September 2022), began serving NASA in 1982, drafting a report on space station habitability in 1986. Still, Roger Wescott opined just six years after Neil Armstrong's epic step on the moon that a true extraterrestrial anthropology with a genuine "ethnological component can hardly be said to be functioning at all until such time as ethnologists begin to observe and describe the nascent cultures and subcultures of human communities in earth orbit, in lunar orbit, or on the lunar surface", not to mention distant planets and star systems [2] (pp. 14–15). That time is no longer so far off.

Meanwhile, the anthropology of space has matured into a legitimate specialty in the discipline. It is possible for college students to take courses in space anthropology, such as Lisa Messeri's offering at the University of Virginia or Kathryn Denning's at York University (both titled "The Anthropology of Outer Space"). Moreover, Savannah Mandel recently authored a short enthusiastic review of the accomplishments of space anthropologists, easily accessible for scholars, students, and the general public [12], and Peter Timko et al. survey the field in relation to their new course at Poland's Jagiellonian University and as members of the Anthropological Research into the Imaginaries and Exploration of Space group [13]. So, the training of the next generation of space anthropologists is underway.

## 3. The Politics of Space Societies

Scholars from many disciplines have commenced the work of imagining and planning the politic organization and governance of space communities, typically employing basic concepts of law and sovereignty; there is even a journal, *Astropoltiics*, in operation since 2003, published by the Astropolitics Institute (www.astropoliticsinstitute.org, accessed on 14 September 2022). Most of this commentary, not surprisingly, falls squarely within conventional political thinking and extrapolates earth law, sovereignty, and nationalism to outer space. For instance, earth governments through the United Nations adopted the Outer Space Treaty (formally, Treaty on Principles Governing the Activities of States in the Exploration and Use of Outer Space, including the Moon and Other Celestial Bodies) in 1967. Obviously, this move was intended to regulate and restrain terran states from monopolizing and misusing space: explicitly (if a bit unrealistically), the treaty insisted that space exploration and development "shall be carried out for the benefit and in the interest of all countries" and that outer space "is not subject to national appropriation by claim of sovereignty, by means of use or occupation, or by any other means" [14]. Further,

spacefaring governments bound themselves to established international law and forbade placing weapons in space or building military bases on celestial bodies. This was followed by the 1979 Moon Treaty (Agreement Governing the Activities of States on the Moon and Other Celestial Bodies); however, by January 2021, only eighteen countries had ratified the latter treaty, including no countries with active space programs [15].

It is glaringly apparent that current space law reproduces and projects earth-bound political norms and practices into space, and it is hard to conceive how it would do otherwise. Exploring, settling, and governing space is pictured as something that earth states will do, no doubt (despite the idealistic language) for their own partisan nationalistic interests. Hence, development scholar Alan Marshall, writing in the journal *Space Policy*, was almost certainly correct when he declared that "the present politico-legal regimes which govern prospective space development (and, moreover, the philosophical inclinations of many of those involved in formulating such regimes) dictate that Solar System development *will be of an imperialistic nature*" [16] (p. 41, emphasis added). First, it will, and so far has been, "undertaken by a few technologically elite space-capable nations who will appropriate the commonly-owned resources of the Solar System for themselves" (p. 49), and it is unlikely that new state or corporate players such as China or Amazon, respectively, will be different.

Second and more profoundly, the race for space does reflect, and always has reflected, a frontier mentality akin to the one characteristic of earthly empire-building from the sixteenth to the twentieth century (as summed up by *Star Treks*'s reference to "the final frontier"). Peter Redfield, who had previously written an ethnography of space projects based in French Guiana [17], concluded two decades ago that "outer space reflects a practical shadow of empire", with the "otherness" of prior drives toward colonization "removed from the globe" and displaced onto sites off-planet; additionally, the penetration and conquest of space expressed the familiar "masculine adventure of earthly colonialism" [18] (p. 795). In short, the "rhetorical link between outer space and colonial history requires little introduction. Anyone with a passing acquaintance of the Space Age is familiar with its frontier metaphors and allusions to European colonial expansion" (p. 796).

The late entry of nongovernmental agents, especially corporations such as the aforementioned SpaceX and Blue Origin, into the space game changes the political equation somewhat. As portrayed in another novel, Mary Doria Russell's 1996 *The Sparrow*, a nongovernmental actor (in Russell's story, the Catholic Church) may for various reasons reach a planet before any government, due to its wealth, freedom from state bureaucracy and public opinion, profit motive, and/or missionary zeal. Law professor Ariél Ferreira-Snyman holds that it "seems inevitable that once a private company has de facto control over a space object such as the moon or an asteroid, such control may become legal once the majority of states recognizes or at least does not object to such appropriation" [19] (p. 27). Space-oriented corporations and organizations may behave, at least temporarily, as patriotic proxies of particular states, but experience has shown that such fealty has limits, temporal and practical. Accordingly, law scholar Melissa Durkee reckons that we can already discern "that private companies are themselves developing the international law of outer space", naturally in a way "that is permissive of appropriation of space resources" [20] (pp. 428–429). Obviously too, no corporations or other private entities were signatories to any space treaties and thus do not feel bound by such pacts.

Meanwhile, some observers have speculated about how new legal principles and structures may be needed for space communities and how—to pivot back to the theme of the present essay—space communities may evolve their own distinct governance systems. As one example, Sara Bruhns and Jacob Haqq-Misra [21] of the Blue Marble Space Institute of Science propose some guidelines and procedures for "sovereignty on Mars". Among their suggestions are the assignment of lots (of up to one hundred kilometers in radius) for each colony and a "planetary park system" that would ban any unapproved habitation or use within its boundaries. They recommend loose and decentralized authority on the red planet, in which each colony chooses its own governance style, overseen simply by a weak "administrative body" headed by a Secretariat and a "temporary tribunal" of representatives

from all Martian colonies (something such as the colonial American Continental Congress). They further advise that Martian politics be self-funded, to avoid dependence on, and interference from, earth governments—effectively beginning to dissolve the political and economic bonds between worlds.

## 4. Discussion: On Space Colonialism, Ethnogenesis, and Exonationalism

Much of this thinking would be idle imagining if not for Project Artemis, an American plan to return to the moon by 2024, to plant a permanent human presence there by the end of the 2020s, and to initiate the reach to Mars and beyond (in the process, the United States has forsaken international treaties in favor of non-binding bilateral or multilateral agreements and understandings) [22]. China also has declared intentions to send manned missions into space and construct its own space station, and India and the European Union have announced their entry into the new space race. There may be multiple off-earth sites for ethnographic research sooner than we expected.

As already suggested, and at the heart of this essay, the journey into space can be and overtly and often enthusiastically has been conceived as a new wave of colonization and of attendant colonialism. More than one anthropologist or other scholar has drawn conspicuous parallels between previous human migrations and the leap into space. In Douglas Vakoch's edited volume *Archaeology, Anthropology, and Interstellar Communication*, Douglas Raybeck advises that "some of the more notorious examples of European and U.S. colonialism may enrich our discussion of the possibilities" of space settlement [23] (p. 144); toward that end, he summarizes the histories of colonization of Central America, northeastern North America, Japan, China, and New Zealand. Of course, much of this discussion revolves around the issue of contact with extraterrestrial beings possessing intelligence and culture (and thus falling outside the purview of conventional anthropology but within the purview of a reinvented anthropology or xeno-anthropology), which is for now a problem of the remote future. Even so, if and when that day arrives, the colonization of space is likely if not inevitable to recapitulate the more unpleasant aspects of past terran colonizations, in one of two ways: humans may attempt to dominate if not eradicate the local alien populations, or we may find ourselves outmatched by a superior species that dominates or eradicates us. Indeed, there is no convincing precedent for a peaceful and mutually beneficial relationship between human colonizer and non-human colonized (in Russell's novel, things go decidedly badly for the human pioneers); hence, the evocative title of American studies scholar M. Jane Young's 1987 essay "Pity the Indians of Outer Space", in which Native American peoples reflect on the invasion of outer space through the lens of the invasion and subordination of their own spaces [24].

Some day we may encounter extraterrestrials, with all of the practical, philosophical, military, "racial", and—if their DNA is sufficiently similar to ours—sexual problems that such an encounter will raise. With or without (and most likely without) native populations, in the initial stages, possibly for the first decades or more, human space bases and settlements will probably continue to be, and consider themselves to be, offshoots of earth nations. Such was the case with the early trading posts and colonies on the coasts of the Americas, Africa, Australia, etc., which persisted in identifying as beachheads of the mother country for many years. Yet, as much as scholars and futurists invoke the paradigm of colonization to contemplate and prepare for space migration, the implications of that paradigm have not been explored fully and to their logical end—an end that anthropology cannot describe in great detail but can envision in general contour, given our prior knowledge and critique of earth colonialism.

Let us begin by noting that here on earth there have been various distinct practices and manifestations of colonization and colonialism. Most basically, we can distinguish between colonies of exploitation, where the main purpose of the colony was resource extraction and where few emigrants from the mother country relocated (leaving the productive labor to natives or imported slaves), and colonies of settlement, where large numbers of emigrants—sometimes enough to outnumber the local population, always enough to subdue and co-opt

them—seized and occupied the territory. This latter, much-maligned "settler colonialism", as theorized by Patrick Wolfe, is fundamentally "premised on displacing indigenes from (or replacing them on) the land" [25] (p. 1), obeying a "logic of elimination".

Again, while classic settler colonialism in space is (presumably) far off at best, there are other lessons that historical and anthropological experience of colonialism can teach us today. Most immediately, the privatization of space exploration and the capitalist inclinations of NewSpace should come as no surprise since both were entirely heralded by Western colonialism. States were often not the first or main colonial actors; rather, private enterprises, often in the form of chartered companies, took the lead in founding and managing colonies, for the glory of the homeland, no doubt, but also for the pecuniary benefit of company officers and shareholders. The Virginia Company, the Plymouth Company, the Hudson's Bay Company, the Royal Niger Company, and most famously the British East India Company and Dutch East India Company were the early-modern equivalents of SpaceX and Blue Origin (as it were, chartered offshoots of their founders' main enterprises, Tesla and Amazon, respectively). States were often content to let entrepreneurs and investors take the risk and absorb the expense. As is well known, institutions such as the British East India Company were more than businesses, functioning essentially as the government and army of colonized territories until the state intervened.

In the opening stages of space colonization, some combination of private/capitalist and state/nationalist forces will certainly drive the process and share the sovereignty. We might regard them as what Martjin Goosensen [22] calls the "space actors" whose agency establishes the first space colonies, no doubt small and fragile affairs such as the coastal footholds of Europeans in Africa or China. Some may fail, similarly to Roanoke in North America. Those that survive, especially in the foreseeable future of near-earth space habitation, will remain what Cameron Smith, in perhaps the most thorough presentation of space anthropology to date, names "terrestrially-tethered settlements", which will "maintain, for several generations at least, substantial (e.g., organizing, structural) biological, cultural, and/or political ties with Earth" [1] (p. 27). As a result of their proximity to their terran source, communication, exchange, and population movement between earthly metropole and celestial periphery will be relatively easy, and the small-but-growing colonies will not be space actors proper. This situation, however, will not last forever, as exocolonization proceeds over time and space. This will be especially the case in non-terrestrially tethered settlements; due to their population growth, the age and evolution of their society, and especially their distance from earth, they will "become structurally independent of Earth, in biological, cultural and political connections", then "we may expect more rapid adaptation and diversification of our biology, culture, and technologies" (p. 27). At some point, these space settlements will become not just earth societies in space but *space societies* and space actors in their own right.

While space migration and colonization are in one sense a recapitulation and extrapolation of the earthly history of migration and colonization, in another sense "settling in space will be a revolutionary act, because leaving Earth to colonize new worlds will change humankind utterly and irreversibly" [26] (p. 166). Yet, to a certain extent the same could be said of those previous terrestrial population movements, which introduced travelers to unknown variations of humanity and, through syncretism and interbreeding, spawned new variations of humanity, as well as previous revolutions such as the agricultural and industrial revolutions, which reshaped humanity and the planet.

The transformation of humankind itself in space, and of our conceptions of "the human", will begin from the very inception of space settlement. Here, physical anthropology and population genetics have valuable insights to contribute. Undoubtedly, space pioneers will be a small group, such as the passengers on the *Mayflower* who settled Massachusetts, and also a highly select and atypical group, not at all representative of the general human population. Almost certainly, it will consist primarily of members of a few earth societies such as the United States, Russia, China, some European countries, and a few others; peoples often neglected or excluded on earth, such as Africans and indigenous

nations, are likely to be neglected and excluded again. However, the "founder effect" of the first space colonies will be much greater. As Mary Oberthur anticipated nearly fifty years ago, these expeditionary groups will likely be "chosen for their professional competence, emotional stability, and reliability of judgment", as well as their general health, probably from a narrow age range [27] (p. 175). They may be physiologically enhanced with various technologies, at the extreme virtually cyborgs who have already crossed a threshold of post-humanness. Further, they will most likely bear other distinct cultural and personality traits. They will be more scientifically oriented than average, therefore perhaps less religious and less artistic; they will almost necessarily be more intelligent, more adaptable, maybe more tolerant and liberal, although they may have more military leanings. If they are recruited by a corporation or other private entity, they may be more capitalistic of mind and more loyal to the company. Either way, they will not constitute a simple transfer of earth culture and the human species in toto into space.

Once this founding population is established on or in orbit around another celestial body, we can expect over time that genetic (or techno-genetic) and cultural drift will occur. Physically, if the group is sufficiently isolated, its unique gene pool and any local adaptations, accidents, and mutations will gradually shift it further away from earthly gene frequencies. Phenotypically, space residents will adapt to the local conditions of atmospheric pressure, gravity, soil chemistry, light and cosmic rays, and such factors. At an even faster rate, cultural changes will appear and accumulate. As we noted, some of these will be necessary and intentional: space colonies will devise local solutions to their economic and political challenges. However, observers have pondered changes in every element of space culture, from marriage and kinship arrangements to language; Smith specifically suggests that "we can expect vocabularies to change, both by dropping use of older words and phrases that no longer 'map to a useful reality' and by the invention of new words and phrases to reflect and order new realities [1] (p. 168), leading to colony-specific dialects.

We should equally expect philosophy, religion, and spirituality to evolve through the generations, eventually producing novel beliefs and practices, similar to the processes that came to separate and alienate the Urrasti and Annarans in Le Guin's novel. Innovative interpretations of terran philosophies and religions will coalesce, leading to site-specific new religions analogous to Mormonism in the United States; subsequently, hybrids of earth-born and space-born, or of two or more space-born, religions would follow, echoing phenomena such as Candomblé, Umbanda, or Santo Daime in the Americas, regarded as strange if not sacrilegious to earthlings and to each other. A preview of what we might expect to see is the "technopaganism" of contemporary Silicon Valley documented by Stef Aupers [28], in which highly rationally minded folks attribute magical or animistic qualities to computer programming, digital technology, and the Internet.

With this awareness, analysts have foreseen "a dramatic cultural rediversification of humankind" [26] (p. 166) that would culminate in what we have recognized since the 1970s as ethnogenesis, "the development, both through environmental conditioning and deliberate collective choice, of new cultures and subcultures" [2] (p. 21). Ethnogenesis or the becoming of a new category or culture of person/human is a well-documented and much-researched topic in anthropology and other social sciences, and we understand that major alterations in cultural identity are possible with fairly minimal alterations in cultural ideation or behavior. A case in point is colonial America or Australia: both sets of settlers came relatively quickly to deem themselves different from their mother country, although their cultures did not diverge so significantly from the source (both, for instance, retained the English language, Christianity, and much of Anglo culture). The same happened in all previous historical colonies, where separation from the homeland and adaptation to local circumstances (including contact and intermarriage with indigenous peoples) launched the process of differentiation of culture and identity.

The upshot here is that, presumably before space colonists actually become objectively different, physically and culturally, from their terran ancestors, they will feel different; here is the point that prior discussions of off-earth politics and nationalism have overlooked. At

first, space settlers will continue to identify with and express loyalty to their earth nations of origin; nationalism in space will be essentially earth nationalisms relocated to other places (i.e., settlers will classify themselves as "Americans on Mars" or "Russians on the moon"), just as the pioneering settlers in the Americas considered themselves Englishmen in New England or Spaniards in New Spain (Mexico), even likely preserving the habit of naming space settlements after terran sites (e.g., New America or New Beijing).However, over time—and it took less than 150 years in the British colonies of America—some or all of the descendants of those first colonists will begin to think of themselves as "Martians" or "Moonians" or, once we reach other star systems, "Alpha Centaurians", etc. Or, if there are multiple Martian colonies, inhabitants may identify even more locally, attached and loyal to their particular settlement or city. At that point, we have true *exonationalism*, affiliation with a society or "nation" that is not earth-based and has no affective ties to any earth state or nation.

As Valentine, Olson, and Battaglia put it, the endpoint of this physical and cultural differentiation will be a population of space-born humans "who regard themselves as 'no longer Earthlings'" [11] (p. 11). Many, similarly to their British counterparts in colonial Boston and New York, may have never visited the mother country/world and have no memories and only limited knowledge of and affection for it. Three consequences are thereby predictable with a fair degree of confidence. First, exonationalistic space societies will begin to clamor for their independence from earth; in fact, if they are millions of miles or even some light-years from earth, they may have minimal and highly delayed contact with the home planet anyhow. Any country or company that thinks it is founding permanently subservient branch-settlements in space is ignoring centuries of earth colonial history. The exocolonies may come to feel exploited and oppressed, by burdensome taxes, foreign and onerous laws and such, or disrespected as mere resource producers with no inherent rights or identity. They may simply claim their prerogative to self-determination, refusing to remain subsidiaries of some earth power. Or unrest may arise from the ambitions of individuals or parties within the colonies. Once begun, unless human technology has invented much faster space flight with much larger ships, the earth could probably not transport an armed force quickly enough to suppress any revolutionary uprising, so political independence and exosovereignty would be relatively easy to achieve.

This raises the second expectation. Barring a deep change in human nature, it is almost inconceivable that rivalries, hostilities, and actual violence would not flare between colonies and the earth and also between disparate colonies. Independence movements themselves would likely feature an amount of conflict and violence, as would political struggles within and between colonies. Larger or more aggressive colonies might launch raids if not conquests against their "neighbors", competing for territory and resources, redrawing boundaries, and annexing other settlements. Indeed, in a sort of Boasian or Geertzian integrative revolution, it might be seen as progress (by some, for some time) to aggregate these separate "tribal" local space identities into grander exo-nations.

Third, experience tells that we may expect the rise of new kinds of space-based imperialism. Some former colonies may choose to enter alliances and treaties with others and with earth states or an eventual global earth government; some may even remain in an Earth Commonwealth. It is even imaginable—since it happened in the intensely nationalistic context of modern earth—that a competing impetus of coexistence and cooperation, even of exocosmopolitanism (perhaps in the form of a United Planets or United Worlds), will emerge, truly realizing the notion of being a citizen of the cosmos. However, when they attain a critical size, former colonies may start to forge their own associations and empires, out of existing colonies or of colonies that they themselves fling into space. In effect, prosperous and independent space societies can become the base from which new colonial forays are sent. An extraterrestrial Monroe Doctrine might ban earth from interfering in space.

One other pernicious colonial process that is likely to be revived in space is "racism" and debates about "the human". European colonialism, according to many critics, exacer-

bated if not created race concepts and racism, as humans of one kind confronted humans of other kinds. We are well aware of literal debates over the humanity of indigenous peoples, as in the sixteenth-century argument between Bartolomé de las Casas and Juan Ginés de Sepúlveda. Technology reporter Caroline Haskins [29] has already contended that space exploration and colonization are infected with racist thought, noting that "there is the risk that the same racist mythology used to justify violence and inequality on earth—such as the use of frontier, 'cowboy' mythology to condone and promote the murder and displacement of indigenous people in the American West—will be used to justify missions to space". A conference on "decolonizing Mars" in June 2018, well in advance of the peopling of Mars and hopefully pre-empting the worst excesses of this colonization, explored "how using a colonialist framework in space reproduces past harm from humanity's history on Earth" and pondered "fresh pathways for thinking about space exploration by stepping away from the ways we usually talk about space, which by definition is 'decolonizing' the topic" [30]. It was followed up in September by a conference on "Becoming Interplanetary: What Living on Earth Can Teach Us about Living on Mars".

What the history of earth has taught us, which anthropologists such as Boas, Melville Herskovits, and particularly Ashley Montagu have bravely combatted, is that humans are keen to notice even small differences in bodies and behaviors and to become intensely if not violently ethnocentric and condescending about them. Earth societies will possibly look down on space societies as inferiors and provincials, similarly to how Britain did toward America and Australia; for their part, space societies might dismiss their earth counterparts as snobs and elitists or as worn-out vestiges of the human past, Neanderthals to their own futurist humanness. Meanwhile, as observable phenotypic differences appear and grow, the peoples of various exoworlds may be categorized, and may identify, as a new "race", a novel line of humans, with all the tensions that such categories entail. At some point far in the future, one or both sides may recognize a speciation (most likely the space-dwellers declaring themselves a new and superior version of celestial Homo sapiens compared to their primitive earth ancestors). As Valentine summarizes it in reference to Martian-humans,

> Some might redefine the species designation *Homo sapiens* (wise human) in locative terms as *Homo terrans* (Earth humans) from the perspective of *Homo ares* (Mars humans). From Mars, Earth will be just another star, and one you could mistake for Jupiter or Betelgeuse; as Giovanni reminds us, it would have to be pointed out to you. It would not be the center from which to fix accounts of the human. Or perhaps others may emphasize not location but the more pressing ontological affordances of the gravity relation, so that those people on Earth and Mars may become distinguished equally as *Homo pondus* (mass-held humans) from the perspective of Island Three's [an imagined "massive rotating cylindrical settlement" permanently suspended in space] *Homo gyrari* (rotation-held humans)" [31] (pp. 202–203).

Melchiorre Masali and various colleagues, for instance, speculate how low or zero gravity environments may compel us to engineer not only our tools (from laptop computers to toothbrushes) but our bodies, to the point where we "may obviously end up as a new species" [32] (p. 176; see also [33]).

Ideally and hopefully, some or most humans would adjust by expanding their concept of "the human" to include emergent physical and cultural types, as some or most early-modern and modern earth colonialists did after they met Native Americans, Australian Aboriginals, and others who challenged their given notions of what "human" means. Unfortunately, as long as racist thinking prevails, strains, inequalities, and conflict may ensue over these real and putative differences. One can even envisage movements to keep earth or Mars "pure" (e.g., "Martians will not replace us") or protests against discrimination and violence toward non-earthlings (e.g., "Alpha Centurian lives matter").

### 5. Conclusions: Anthropology Post-Earth and Post-Human

If this admittedly speculative but disciplined application of anthropological knowledge and concepts to the project of space colonization has accomplished anything, it is to disrupt "the assumption (whether rightist or leftist) that the encounter with space will simply produce a repetition, extension, or logical conclusion of history, human sociality, exchange relations or any other human phenomena that have emerged on the surface of our planet" [10] (p. 1063). Or, more accurately, human space societies will not be mere copies or transplants of earth societies and relations, any more than Western colonies in Africa, America, and Asia were clones of Western cultures and institutions. Surely we understand that, as much as colonizers might attempt to export their societies intact and unchanged to another location, those colonies will develop cultures, histories, institutions, and identities of their own. From the perspective of early-modern European countries, the United States, Canada, Australia, Mexico, Kenya, and all of the other "new states" are earthly examples of exonationalism, nationalism outside of the range and precedents of the past. This anti-colonial, self-determining terran exonationalism was unexpected and unwelcome for Western colonizers, but today we are in a better position to anticipate and prepare for it.

Since it will be years, if not decades and centuries, before there are extraterrestrial "sites" for participant-observation, the anthropology of space colonialism and exonationalism is an anthropology of the future. Not so long ago, many pundits would have judged anthropology unfit for such a task. Anthropology was defined, even by some of its leading figures such as Radcliffe-Brown, as "the study of what are called primitive or backward peoples" [34] (p. 2) and thus thoroughly married to "tradition" and the past. If this was ever true—and it really was not—it has long since ceased to be the case. As Victor Buchli insists, anthropology has been preparing for this moment, particularly through its prior earth-bound "establishment of multi-sited methodological approaches and the emergence of online ethnography in digital realms", which emphasize *co-presence* even if not actual physical *co-location* [35] (p. 29). Moreover, anthropological methods will continue to evolve in extraterrestrial settings, including ironically, if Buchli is correct, a revival of "armchair anthropology" or anthropology-at-a-distance. We may also depend on reports from the non-professional "man on the spot" until professional anthropologists arrive in space; one can even imagine a cosmic "Notes and Queries" guidebook for such a purpose!

Space colonization will provide new material for ethnographic description and comparison, but it will also achieve more: similarly to totems for Lévi-Strauss, space colonies will be good to think. They will, as already prophesied, stretch our conception of what it means to be human, as biological, technological and cultural evolution continue and accelerate in increasing isolation from terran populations and societies, as part of our increased attunement to "emergent forms" [36], forms not only of culture but of humanity itself. If and when humans interbreed with extraterrestrial species, as imagined in yet another science fiction series, Octavia Butler's *Xenogenesis*, with its anthropologist main character, the problem will become most acute. That much has happened before, as early-modern Western explorers met people of other colors and cultures, although the scale and consequences would be greater. Fortunately, anthropology has aleady begun to contemplate the post-human too [37,38]. Indeed, Alan Smart and Josephine Smart maintain that we have always been post-human, if by that term we mean co-evolving with our physical and cultural environments [39]. Obviously, then, post-humanism is not only our species' future but its past and its very nature, as evinced and only accentuated by the ever-closer integration of technology and human bodies (from ubiquitous cell phones to medical devices and computer chips implanted in our flesh. The future of "cyborgs" (cyber/organisms) is here, as is a cyborg anthropology at least since Downey, Dumit, and Williams' 1995 paper by that name [40]; see also [41,42].

Even more, with space colonization and the appearance of true off-earth exonationalism, there will transpire a necessary and salutary "deterrestrialization of anthropological thought" [43] (p. 207). The effects would be far-reaching; just as post-colonial scholars on

earth strive for a decentering and provincializing of the West, post-exocolonial scholars will strive for a provincialization of the earth itself. Humanity would no longer be a one-planet, earth-only species; more, as exonationalisms crystallize and strengthen, the earth would become increasingly peripheral to them (in the way that Europe became peripheral first to the United States and Australia and then to Africa, Asia, etc.). In a regime of robust exonationalisms, humans on or around other heavenly bodies, or undertaking multigenerational space voyages, would no longer call themselves "earthlings" although they might still call themselves "humans", or might not. As Messeri grasps, the "physical dislocation" of humans from the earth and their physical, psychological, and cultural orientation to space "would change what it means to be human as it would no longer be in reference to existence on Earth" [8] (p. 194). At the extreme, the earth might diminish to a resource-exhausted and/or uninhabitable planet (due to pollution, global warming, and/or nuclear war), or at the very worst it might be destroyed by a cosmic collision (all reasons why Elon Musk for instance is keen to launch humans toward other planets). Indeed, political scientist Nicole Sunday Grove goes so far as to opine that the unlikely space program of the United Arab Emirates, dubbed "Mars 2117", is a deliberate attempt to escape and abandon the earth and its territorial states, a "vision of exodus" [44]. (To see a promotional video on the project, visit www.youtube.com/watch?v=oNNBBHeuoXw, accessed on 14 September 2022).

It is evident that the movement of humanity off their only current home world, and the attendant and inevitable ethnogenesis of alternative identities and nationalisms, will have weighty implications for humankind and for anthropology, the science of humankind (which will have been, until then, a science of humans on earth exclusively, an "earthropology"). Stefan Helmreich goes so far as to forecast a new kind of "extraterrestrial relativism"—particularly if we ever do find other intelligent and cultural species in space—which would be at least a non-earth-centric relativism and potentially "a non-anthropocentric relativism in which humans (as well as other creatures, and, at its limits, life itself) may be entirely absent" [45] (p. 1126–1127). Such issues and unprecedented modes of thought and being lie far in the future, but anthropology can and must commence reflection on them now. For today, this essay and the literature and discoveries on which it is based, as Valentine aptly says, "are all terrestrial first readings that should be the starting point of our thinking about humanness in space, not its conclusion" [30] (p. 204). Moreover, similarly to Western anthropologists in the twentieth and twenty-first centuries, it is not for us to tell "others" how to think and what to be; rather, people residing in and committed to their extraterrestrial worlds—those future exonationalists—will

> have their own critical theories of the universal and specific, of difference and relation, of the human and nonhuman, or of (what on Earth may be distinguishable as) history, politics, ethics, kinship, Indigeneity, species, race, or society ahead of their framing in terrestrial terms, of whatever variety: critical, postcolonial, Indigenous, settler, or entrepreneurial (pp. 203–204).

It is with humility that we will hopefully accord them humanity, insofar as they seek it, and use our scholarly tools to enhance understanding, respect, and acceptance.

**Funding:** This research received no external funding.

**Institutional Review Board Statement:** Not applicable.

**Informed Consent Statement:** Not applicable.

**Data Availability Statement:** Not applicable.

**Conflicts of Interest:** The author declares no conflict of interest.

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
