# Peer review of "Space Colonization and Exonationalism: On the Future of Humanity and Anthropology"

_humans, doi:10.3390/humans2030010_

Round 1

Reviewer 1 Report

Space Colonization and Exonationalism: On the Future of Humanity and Anthropology

Review

The manuscript addresses a very important and relevant topic for the future of humanity beyond the earth and the study thereof. It serves as a timely call or reminder for anthropologists to continue engaging in forward thinking and work towards refining possible anthropological frameworks for analyzing future space migrations and their potential implications. The paper explains how and what anthropologists have so far written/done on space colonization and what is still outstanding (lines 48 - 60). It uses current and historical evidence to point out how anthropology can position itself as humanity slowly readies itself for future space occupation. Drawing from this evidence and some fundamental anthropological concepts like differentiation, the author is able to convincingly explain such likely consequences of space colonization as ethnogenesis, political/cultural tensions, continued inequalities, etc., while emphasizing that ´´human space societies will not be mere copies or transplants of earth societies and relations, {} ´´ (lines 473-474). Such informed analysis, and forward thinking is a positive contribution to the field of anthropology and outer space studies.

The paper is written with great logic and clarity. I like how the author leaves no room for ambiguity as they explain what the terms they use in the paper mean, for example “exonationalism” (lines 56-59), ´´ethnogenesis´´ (lines 353-355). The author demonstrates extensive knowledge of the topic and has an excellent way of weaving together the past, the present, and the future with such ease that the reader is kept engaged and left thoroughly informed about the contributions of anthropology ´´ {} toward understanding the present and future of space colonization´´ (line 10). There is in general great engagement with literature and a good balance in the use of sources from different fields and disciplines.

One point that the author could perhaps consider adding is that some universities already offer courses on the anthropology of outer space, for example, https://cte.virginia.edu/anthropology-outer-space, https://www.yorku.ca/laps/anth/course-description/anth-3270-3-0-the-anthropology-of-outer-space/ - such courses can be used to further develop ideas on how anthropology can contribute towards explaining the current and future of space colonization and its impact on humanity as well as future space societies.

Overall, the paper raises interesting arguments that are qualified and supported by convincing evidence.

Author Response

Thank you very much for this highly positive review. I will certainly add a mention of university courses on the anthropology of space, as that greatly strengthens the argument that space is a legitimate and important subject for anthropology. I will even look at their syllabi if I can to determine what topics they cover in such classes.

Reviewer 2 Report

Very interesting and "visionary" article, broadening (literally) the horizons of anthropology.  I leave only 2 comments and some bibliographic suggestions that may be of interest:

- Considering the harsh conditions of most known extraterrestrial contexts, it is likely that future human settlers will have to be equipped with technology that will certainly challenge the boundaries of what it is to be human. It would be interesting (if possible) to develop a bit more the analysis on the issue of post-humanism as a plausible scenario associated with space colonization.

- The author put a lot of emphasis on the possible extraterrestrial dynamics of creating borders and boundaries (exonationalism). What about the possibility of border-crossing dynamics (“exotransnationalism”; “exocosmopolitanism”)?  

- Bibliographic suggestions:

https://doi.org/10.1080/19428200.2021.1973332

https://iris.unito.it/bitstream/2318/135452/1/Micheletti%201187126%20di%20UGOV_bozza.pdf

https://doi.org/10.1007/s00779-010-0324-6

https://doi.org/10.1111/1467-8322.12726

https://static1.squarespace.com/static/54d7c6f0e4b081035491f527/t/61e09116cc04fc38c73a85dd/1642107161461/Welcome+to+Mars+space+colonization+anticipatory+authoritarianism+and+the+labour+of+hope.pdf

Author Response

Thank you for your positive review of my article. I think the points you raise are valid and important, and I will try, within the limitations of word count, to include some additional discussion of post-humanism. I think that the point about exocosmopolitanism is interesting and worthwhile, and although I expect that humans will be more exclusive than inclusive in space (since they are so on earth!), I see the value in mentioning the possibility as an alternative current in space settlement, just as it is an alternative current to nationalism on earth.

Reviewer 3 Report

This is an excelent paper. Timely and academically impeccably structured

Author Response

Thank you very much for your succinct and positive review. It is not often that a colleague does not have something critical to say!